# Clinical Outcomes and Prognostic Factors in Nonmetastatic Castration-Resistant Prostate Cancer Treated with Androgen Receptor Signaling Inhibitors Therapy

**DOI:** 10.3390/cancers16152659

**Published:** 2024-07-26

**Authors:** Ryo Fujiwara, Shinya Yamamoto, Kosuke Takemura, Takeshi Yuasa, Noboru Numao, Tomohiko Oguchi, Yosuke Yasuda, Yusuke Yoneoka, Junji Yonese

**Affiliations:** Department of Genitourinary Oncology, Cancer Institute Hospital, Japanese Foundation for Cancer Research, 3-8-31 Ariake, Koto-ku, Tokyo 135-8550, Japan; shinsaku0329@gmail.com (S.Y.); kosuke.takemura@jfcr.or.jp (K.T.); takeshi.yuasa@jfcr.or.jp (T.Y.); noboru.numao@jfcr.or.jp (N.N.); tomohiko.oguchi@jfcr.or.jp (T.O.); yosuke.yasuda@jfcr.or.jp (Y.Y.); yusuke.yoneoka@jfcr.or.jp (Y.Y.); jyonese@jfcr.or.jp (J.Y.)

**Keywords:** nonmetastatic castration-resistant prostate cancer, androgen receptor signaling inhibitors, overall survival, prostate-specific antigen, real-world clinical practice

## Abstract

**Simple Summary:**

We retrospectively evaluated the clinical outcomes and prognostic factors of 127 patients with nonmetastatic castration-resistant prostate cancer (nmCRPC) treated with first-line androgen receptor signaling inhibitors (ARSI) in real-world clinical practice. Overall survival (OS), metastatic-free survival (MFS), and prostate-specific antigen–progression-free survival (PSA–PFS) from the initiation of ARSI were assessed. The median OS and MFS for all patients were 79.0 and 42.0 months, respectively. Median PSA–PFS varied among patients treated with enzalutamide, abiraterone acetate, apalutamide, and darolutamide (27.0, 20.0, 10.0, and 14.0 months, respectively; *p* = 0.33). Multivariate analysis indicated that a baseline PSA level of ≥3.67 ng/mL (*p* = 0.002) was significantly associated with poorer OS prognosis. ARSI demonstrated favorable efficacy in nmCRPC patients, with no significant differences in clinical outcomes among different types of ARSI. High baseline PSA of ARSI was found to have a significantly poor prognosis associated with OS.

**Abstract:**

We conducted a retrospective evaluation of the clinical outcomes and prognostic factors in patients with nonmetastatic castration-resistant prostate cancer (nmCRPC) treated with first-line androgen receptor signaling inhibitors (ARSI) in real-world clinical practice in Japan. Between 2012 and 2023, a total of 127 consecutive patients with nmCRPC received ARSI treatment. Overall survival (OS), metastatic-free survival (MFS), and prostate-specific antigen–progression-free survival (PSA–PFS) from ARSI initiation were assessed using the Kaplan–Meier methodology. Clinical factors associated with OS in nmCRPC were analyzed using the Cox proportional hazards model. Among the patients, 72, 26, 12, and 17 received enzalutamide (ENZ), abiraterone (ABI), apalutamide (APA), and darolutamide (DARO) as first-line therapy. The median OS and MFS for all patients were 79.0 and 42.0 months, respectively. Median PSA–PFS was 27.0, 20.0, 10.0, and 14.0 months for patients treated with ENZ, ABI, APA, and DARO, respectively (*p* = 0.33). Multivariate analysis revealed that a baseline PSA level ≥ 3.67 ng/mL at ARSI initiation was significantly associated with poorer OS (*p* = 0.002). ARSI demonstrated favorable efficacy in nmCRPC patients. There were no significant differences in clinical outcomes among different types of ARSI therapy for nmCRP. Elevated baseline PSA at ARSI initiation was significantly associated with poorer OS.

## 1. Introduction

Prostate cancer remains the second most common malignancy among men worldwide [1]. In localized prostate cancer, primary treatments typically include radical prostatectomy or radiation therapy. However, recurrence occurs in 15 to 30% of treated patients [2,3], who subsequently undergo androgen deprivation therapy (ADT) as a standard treatment [4]. Unfortunately, some patients progress to nonmetastatic castration-resistant prostate cancer (nmCRPC), which is associated with increased risks of metastasis and mortality. Median overall survival (OS) and bone metastasis-free survival in nmCRPC patients treated with ADT have been reported as 43.5 and 31.5 months, respectively [5]. 

Recently, three phase III trials—SPARTAN, ARAMIS, and PROSPER—demonstrated that apalutamide (APA), darolutamide (DARO), and enzalutamide (ENZ) significantly extend metastatic-free survival (MFS) and OS for patients with nmCRPC [6,7,8]. The NCCN guidelines recommend offering androgen receptor signaling inhibitors (ARSIs) to nmCRPC patients if their PSA doubling time (PSADT) is ≤10 months [9]. Japanese public health insurance covers ENZ, APA, DARO, and abiraterone acetate (ABI) with prednisone for nmCRPC. Real-world clinical practice shows that PSA progression-free survival (PSA–PFS) and MFS are significantly improved with ENZ and ABI compared to first-generation antiandrogen therapy like flutamide [10]. Novel ARSIs have been associated with better oncological outcomes in nmCRPC patients compared to those not receiving ARSIs [11]. However, studies on the efficacy and safety of ARSIs in nmCRPC patients and comparisons among ENZ, APA, DARO, and ABI in real-world clinical settings remain limited. Furthermore, while some reports suggest PSADT as a prognostic factor for nmCRPC, the optimal cutoff value remains unclear [5,12]. Therefore, additional effective biomarkers are needed.

In this study, we conducted a retrospective evaluation of the clinical outcomes and prognostic factors among patients with nmCRPC treated with first-line ARSIs in real-world clinical practice in Japan.

## 2. Methods 

### 2.1. Study Population

We retrospectively reviewed consecutive patients diagnosed with nmCRPC who were treated at our institution between March 2012 and August 2023. Radiological data from computed tomography (CT), magnetic resonance imaging (MRI), or bone scintigraphy, as well as clinical information from medical records, including physical and pathological examination findings and laboratory results before and during ARSI therapy, were collected based on the attending physician’s records. This study was approved by the institutional ethics committee at the Cancer Institute Hospital, Japanese Foundation for Cancer Research. Written informed consent for ARSI therapy was obtained from all patients before initiating treatment. 

### 2.2. Treatment and Follow-Up Examination 

All patients received androgen deprivation therapy (ADT) throughout the treatment period. PSA progression was defined as a greater than 25% increase in PSA levels over two consecutive measurements separated by at least 1 week [13] despite maintaining low serum testosterone concentrations below 50 ng/mL with ADT and confirmed absence of distant metastases using CT, MRI, or bone scintigraphy prior to the initiation of ARSI therapy. Patients with a history of distant metastases were excluded. ENZ, APA, and DARO were administered every 4 weeks. ABI was administered every 2 weeks for the first 3 months then every 4 weeks after that. The recommended dosages for ENZ, APA, ABI, and DARO were 160 mg/day, 240 mg/day, 1000 mg/day, and 1200 mg twice daily, respectively. Clinical information, including physical examination findings, Karnofsky Performance Status, and laboratory results before and during ARSI therapy, was collected based on the attending physician’s decision. Evaluation for distant metastases was performed using conventional imaging modalities such as CT, MRI, or bone scintigraphy, as determined by the attending physician.

### 2.3. Statistical Analysis

Descriptive statistics for continuous variables are presented as the median and interquartile range (IQR), while categorical variables are reported as frequencies and percentages. PSA–PFS, MFS, and OS were defined as the time from initiation of ARSI therapy to PSA progression, the first evidence of distant metastasis on conventional imaging, and death from any cause, respectively. PSA response was defined as the proportion of the lowest PSA level relative to baseline PSA at the initiation of ARSI therapy. PSA progression was defined as a ≥25% and ≥2 ng/mL increase from the PSA nadir in men who had a reduction in PSA after ARSI initiation and a ≥25% and ≥2 ng/mL increase from the PSA on the date of ARSI initiation in men with no reduction in PSA after ARSI initiation. Data were censored on the date of the last PSA measurement during ARSI treatment [13]. Patients who remained alive without disease progression and those lost to follow-up were censored at the time of their last follow-up or contact. 

PSA–PFS, MFS, and OS curves were estimated using the Kaplan–Meier method, and differences were analyzed by the log-rank test. In addition, we investigated the association between OS and several clinicopathological factors, including Gleason pattern 5 at prostate biopsy, lactate dehydrogenase (LDH), alkaline phosphatase (ALP), PSA at initiation of ARSI therapy, time to CRPC, prior docetaxel chemotherapy, presence of regional lymph nodes, treatment type (surgery, radiation therapy, or no local treatment with combined androgen blockade), and PSADT calculated using the Pound et al. method [14]. The optimal cutoff values for ALP, LDH, time to CRPC, and PSA were determined based on receiver operating characteristic (ROC) curves, and patients were stratified into higher and lower groups based on these cutoffs.

Univariate and multivariate Cox proportional hazard models were used to assess the significant associations between OS and the clinical factors. Hazard ratios (HRs) and 95% confidence intervals (CIs) were calculated to quantify these associations. Statistical analyses were performed using JMP software version 13.0 (SAS Institute, Cary, NC, USA), and *p* values < 0.05 were considered statistically significant.

## 3. Results

### 3.1. Patient Characteristics

During the study period, 193 patients received ARSI therapy for nmCRPC. Among them, 19 patients with distant metastases at the initial diagnosis and 47 patients who received ARSI therapy as the second or later line therapy for nmCRPC were excluded from the study. After excluding ineligible patients, a total of 127 patients who received ARSI therapy as the first-line treatment for nmCRPC comprised the cohort for this study.

The median follow-up period was 37.0 months (IQR, 18.0–61.1) after initiation of ARSI therapy. Disease progression resulted in the deaths of 23 patients (18.1%), while 12 patients (9.4%) died from disease progression and other causes. Distant metastases were evaluated in 109 patients using CT or bone scintigraphy. Patient characteristics are summarized in Table 1. Docetaxel chemotherapy was administered to six patients (4.7%) for the treatment of nmCRPC. The number of patients with Gleason Score (GS) ≥ 8, clinical T stage (cT) ≥ 3, and PSADT < 6 months were 87 (69%), 82 (65%), and 99 (78%), respectively. Local treatment included surgery in 57 patients (45%) and radiation therapy in 46 patients (36%), while 24 patients (19%) did not undergo local treatment. ARSI therapy consisted of DARO in 17 patients (13%), APA in 12 patients (10%), ABI in 26 patients (20%), and ENZ in 72 patients (57%).

### 3.2. Efficacy of ARSI Therapy

During the follow-up period, 69 patients (54%) experienced PSA progression. The median PSA–PFS, 1-year, 3-year, and 5-year PSA–PFS for all 127 patients were 21.0 months, 61.9%, 33.8%, and 23.8%, respectively (Figure 1). A total of 62 (86.1%), 15 (57.7%), 9 (75.0%), and 12 patients (75.0%) achieved a ≥50% decrease in PSA, and 34 (47.2%), 11 (42.3%), 5 (41.7%), and 3 patients (17.6%) achieved a ≥90% decrease in PSA among patients treated with ENZ, ABI, APA, and DARO, respectively (Figure 2). 

The median PSA–PFS and 1-year PSA–PFS rates with ENZ, ABI, APA, and DARO were 27.0, 20.0, 10.0, and 14.0 months, and 63.7%, 64.9%, 50.0% and 60.5% (*p* = 0.33), respectively (Figure 3). The median MFS, 1-year, 3-year, and 5-year MFS rates were 42.0 months, 79.2%, 60.9%, and 30.4%, respectively (Figure 4). The median OS and 1-year, 3-year, and 5-year OS rates were 79.0 months, 99.1%, 81.5%, and 64.4%, respectively (Figure 5).

### 3.3. Safety of ARSI Therapy

Forty-one patients (32%) experienced adverse events (AEs), with 4 (3.1%) classified as severe AEs (≥Grade 3), as detailed in Table 2. AEs were reported in 23 (31.9%), 6 (23.1%), 9 (75.0%), and 2 (17.6%) patients treated with ENZ, ABI, APA, and DARO, respectively. Although APA showed the highest proportion of AEs among all ARSIs (*p* = 0.005), there was no significant difference in severe AEs (*p* = 0.531). 

The most common AEs associated with ENZ, ABI, APA, and DARO were fatigue (n = 11, 15.3%), fatigue (n = 2, 7.7%), rash (n = 6, 50.0%), and fatigue (n = 2, 11.8%), respectively. There was no significant difference in the proportion of fatigue among patients treated with ENZ, ABI, and DARO (*p* = 0.566). Eighteen patients (14.2%) discontinued ARSI therapy due to AEs (ABI: n = 5, APA: n = 4, DARO: n = 2, and ENZ: n = 7).

### 3.4. Predictors of Prognosis in ARSI Therapy

We investigated possible prognostic factors in ARSI therapy based on pre-treatment variables. According to the ROC curve, the optimal cutoff values for LDH, ALP, time to CRPC, and PSA were 192 U/L, 271 U/L, 54 months, and 3.67 ng/mL, respectively. In multivariate analysis, PSA ≥ 3.67 at the initiation of first-line ARSI therapy emerged as the strongest predictor of poor prognosis for OS (*p* = 0.002; HR: 2.93, 95% CI: 1.46, 6.26) (Table 3).

When patients were stratified based on PSAs, differences were observed in the OS curves between those with PSA < 3.67 and PSA ≥ 3.67 (Figure 6, *p* = 0.002). The median OS was not reached for patients with PSA < 3.67, with a 5-year OS rate of 80.9%. For patients with PSA ≥ 3.67, the median OS was 60.0 months, with a 5-year OS rate of 48.0%.

## 4. Discussion

This study of nmCRPC patients receiving ARSI revealed three key findings. First, ARSI demonstrated favorable efficacy for nmCRPC patients. Second, there were no significant differences in clinical outcomes among different types of ARSI therapy for nmCRPC. Third, a high baseline PSA at the initiation of ARSI was significantly associated with a poor prognosis for OS.

Recent three phase III clinical trials, namely SPARTAN, ARAMIS, and PROSPER, have demonstrated that ARSIs significantly improve OS and MFS in nmCRPC [6,7,8]. The phase Ⅱ IMAAGEN study demonstrated that ABI resulted in a significant ≥50% PSA reduction in nmCRPC [15]. Real-world settings have also shown that ENZ and APA are effective and safe, with nmCRPC patients starting ENZ treatment in Japan experiencing a median time to PSA progression of 27.0 months [16] and an 86% PSA ≥ 50% reduction reported in nmCRPC patients initiating APA treatment [17].

Recent multi-institutional real-world studies have indicated that ARSI treatment, including ENZ, APA, DARO, and ABI, significantly improves oncological outcomes compared to conventional hormonal treatments for nmCRPC patients [11]. Our single-institutional study also supports the favorable oncological outcomes of novel ARSIs in real-world clinical settings. 

A systematic review and network meta-analysis focusing on OS and AEs associated with APA, ENZ, and DARO ranked DARO as the most effective OS benefit related to ADT, followed by ENZ and APA, in that order. DARO also showed the most favorable profile regarding Grade ≥ 3 AEs [18]. There were no significant differences in PSA–PFS among the four novel ARSIs (APA, ENZ, DARO, ABI) in our cohort. While more patients experienced AEs with APA compared to the other ARSIs, the proportion of severe AEs did not significantly differ among all ARSIs. 

A multivariate network meta-analysis suggested that AEs associated with novel ARSI therapy do not significantly differ, except for ENZ being ranked as more toxic regarding hypertension and headache [19]. However, the comparative oncological outcomes and safety profiles among ARSIs for nmCRPC patients remain unclear. Therefore, additional studies involving larger cohorts of nmCRPC patients treated with ARSIs are needed. 

Some risk factors affecting metastasis and survival outcomes in nmCRPC patients have been analyzed. Higher PSA concentrations at diagnosis, shorter PSADT, higher Gleason Score, a history of primary intervention, and a shorter interval from ADT initiation to the diagnosis of CRPC have all been associated with shorter time to metastasis [20]. Another study found that metastasis was associated with higher PSA levels at diagnosis, nadir PSA after ADT, rapid ALP rise, and shorter PSADT [21]. Shorter bone metastatic-free survival was observed as PSADT decreased below 8 months [22]. Trials like SPARTAN, PROSPER, and ARAMIS used 6 months as a cutoff point for PSADT in subgroup analyses [6,7,8]. However, while PSADT < 3 months was associated with the highest risk of metastasis and poorer survival outcomes, the optimal PSADT cutoff for predicting oncological outcomes and risk stratification in nmCRPC patients remains unclear [12]. 

ALP has also been suggested as a significant prognostic marker for MFS in nmCRPC patients, although these patients were treated with not only ARSIs but also antineoplastic therapies such as docetaxel and cabazitaxel. Elevated ALP is thought to indicate pre-existing activated micro-metastases [23]. Our study indicated that a baseline PSA ≥ 3.67 ng/mL at the initiation of first-line ARSI for nmCRPC patients was the strongest predictor of poor OS in multivariate analysis. Previous reports have associated baseline PSA with survival outcomes [5,24,25,26], as PSA levels reflect tumor volume. High baseline PSA and a short PSADT in nmCRPC may be the most important factors in predicting progression to metastatic disease and reduced OS [20,22,25,26,27]. Therefore, preventing or delaying metastatic progression using these prognostic biomarkers should be a primary therapeutic goal for treating nmCRPC [28]. To the best of our knowledge, this is the first report to demonstrate that higher baseline PSA levels are associated with poorer survival outcomes in nmCRPC patients treated with ARSIs. Baseline PSA level as a risk factor has been previously reported in nmCRPC patients undergoing ADT therapy.

We acknowledge several limitations in our study. First, it was retrospective with a relatively small cohort from a single institution, and the number of groups other than ENZ was also relatively small, potentially introducing bias. The small sample size may have impacted the robustness of the results, especially in comparing efficacy and safety among different ARSI therapies. Additionally, comorbidity with all nmCRPC patients, which might affect the efficacy and safety of ARSI, was also unclear. Moreover, our cohort included patients diagnosed with nmCRPC based on PSA levels ≤ 2.0 ng/mL at the start of ARSI therapy, despite the conventional definition of CRPC requiring a PSA rise of ≥2.0 ng/mL [15]. In our real-world setting, PSADT took precedence over absolute PSA levels in 31 patients (24%). This inclusion of patients with baseline PSA < 2.0 ng/mL might have influenced survival outcomes and prognostic factors. Lastly, we defined nmCRPC using conventional imaging like CT and bone scintigraphy. Whole-body diffusion-weighted MRI (WBMRI) has been reported to have better sensitivity and similar specificity compared to bone scintigraphy completed with targeted X-rays (sensitivity: 98–100% vs. 86%, specificity: 98% vs. 98–100%) for detecting bone metastases [29]. Additionally, another retrospective study indicated that prostate-specific membrane antigen (PSMA)-PET showed positivity in 196 out of 200 patients with nmCRPC, revealing pelvic disease in 44% and metastatic disease in 55% of patients despite negative conventional imaging [30]. This suggests that some nmCRPC patients may have already progressed to metastatic CRPC detectable by PSMA-PET scan or WBMRI. However, the availability of WBMRI remains limited, and PSMA-PET scans have not yet been introduced in Japan. Therefore, external validation will be necessary after the introduction of these new imaging techniques in the future.

## 5. Conclusions

ARSI demonstrated favorable efficacy in nmCRPC patients. There were no significant differences in clinical outcomes among different types of ARSI therapy for nmCRPC. High baseline PSA at ARSI initiation was significantly associated with poor prognosis for OS. Given the preliminary nature of our small study, further investigations are necessary.

## Figures and Tables

**Figure 1 cancers-16-02659-f001:**
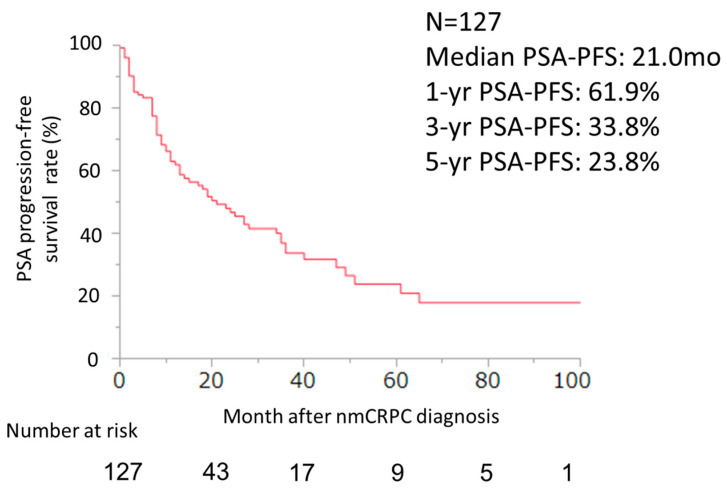
PSA–PFS curve of ARSI for all patients after nmCRPC diagnosis was evaluated using the Kaplan–Meier method (n = 127).

**Figure 2 cancers-16-02659-f002:**
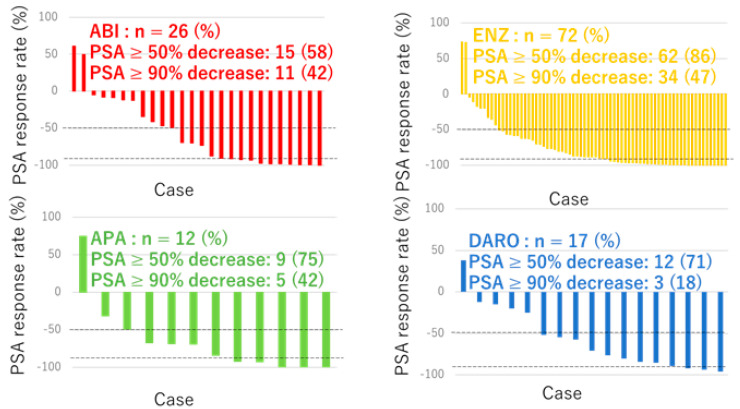
Waterfall plots of the PSA response to ARSI (yellow: ENZ [n = 72], red: ABI [n = 26], green: APA [n = 12], blue: DARO [n = 17]).

**Figure 3 cancers-16-02659-f003:**
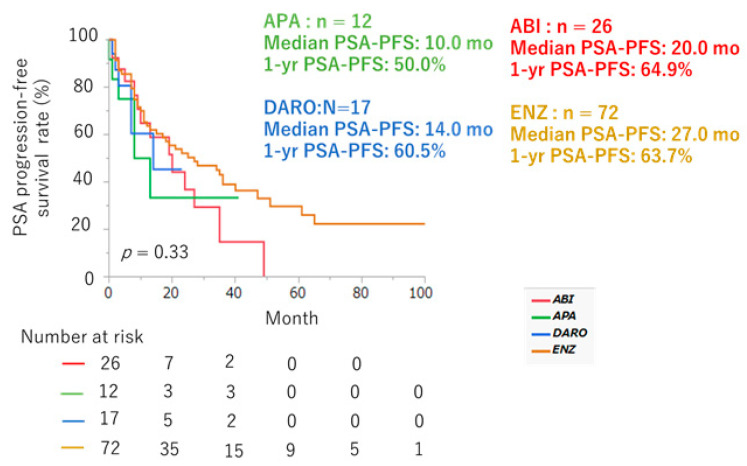
PSA–PFS curve in nmCRPC patients divided by type of ARSI (yellow: ENZ [n = 72], red: ABI [n = 26], green: APA [n = 12], blue: DARO [n = 17]).

**Figure 4 cancers-16-02659-f004:**
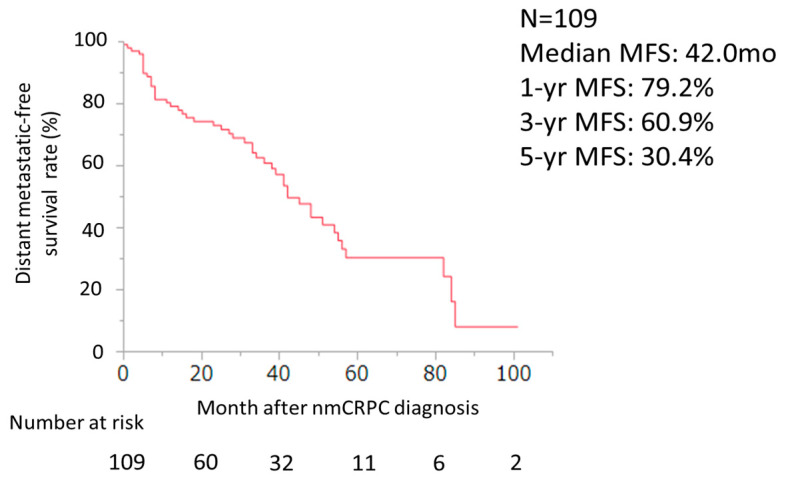
MFS curve of ARSI for all patients after nmCRPC diagnosis was evaluated using the Kaplan–Meier method (n = 109).

**Figure 5 cancers-16-02659-f005:**
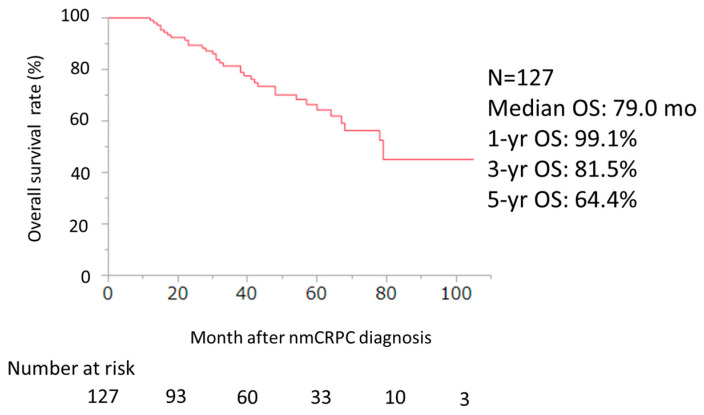
OS curve of ARSI for all patients after nmCRPC diagnosis was evaluated using the Kaplan–Meier method (n = 127).

**Figure 6 cancers-16-02659-f006:**
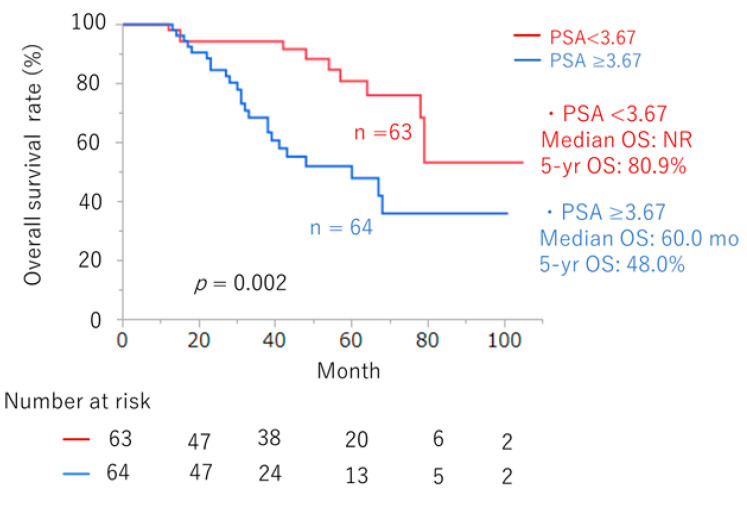
OS curves divided by baseline PSA at the initiation of ARSI ≥ 3.67 (n = 64) and <3.67 (n = 63) ng/mL (*p* = 0.002).

**Table 1 cancers-16-02659-t001:** Patient characteristics (n = 127).

Clinical Factors		Number
Median age at the time of CRPC, years (IQR)		76 (70–81)
Median initial PSA, ng/mL (IQR)		17 (10–59)
Gleason Score (%)	<8	40 (31)
	≥8	87 (69)
Clinical T Stage	1–2	44 (35)
	3–4	82 (65)
	Unknown	1 (1)
Clinical N Stage	0	103 (81)
	1	24 (19)
Local therapy (%)	Surgery	57 (45)
	Radiation	46 (36)
	No local treatment	24 (19)
Previous docetaxel therapy (%)	Yes	6 (5)
	No	121 (95)
PSA doubling time, month (%)	<6	99 (78)
	≥6	27 (21)
	Unknown	1 (1)
First-line ARAT agent (%)	Darolutamide	17 (13)
	Apalutamide	12 (10)
	Abiraterone	26 (20)
	Enzalutamide	72 (57)

CRPC: castration-resistant prostate cancer, IQR: interquartile range, PSA: prostate-specific antigen, ARAT: androgen receptor-axis-targeted.

**Table 2 cancers-16-02659-t002:** Adverse events for each ARSI agent (n = 127).

	ENZ (n = 72)		ABI (n = 26)		APA (n = 12)		DARO (n = 17)	
	Any Grade (%)	≥Grade 3 (%)	Any Grade (%)	≥Grade 3 (%)	Any Grade (%)	≥Grade 3 (%)	Any Grade (%)	≥Grade 3 (%)
Rash	1 (1)	0	0	0	6 (50)	1 (4)	0	0
Neuropathy	2 (3)	0	0	0	0	0	0	0
Fatigue	11 (15)	1 (2)	2 (8)		2 (17)	0	2 (12)	1 (6)
Decreased appetite	4 (6)		0	0	0	0	0	0
Nausea	1 (1)		0	0	1 (8)	0	0	0
Dysesthesia	2 (3)		0	0	1 (8)	0	0	0
Dysgeusia	1 (1)		1 (4)		1 (8)	0	0	0
Edema	1 (1)		1 (4)		0	0	0	0
Dizziness	2 (3)		0	0	0	0	0	0
Neutropenia	1 (1)		0	0	0	0	0	0
Hepatic dysfunction	0	0	2 (8)		0	0	1 (6)	0
Hypokalemia	0	0	0	0	0	0	0	0
Hypertension	0	0	1 (4)	1 (4)	0	0	0	0

**Table 3 cancers-16-02659-t003:** Predictors of overall survival in nmCRPC patients treated with ARSI therapy.

Variables		Univariate		Multivariate	
		HR (95% CI)	*p* Value	HR (95% CI)	*p* Value
Gleason pattern 5 at biopsy	Yes vs. No	1.75 (0.89, 3.56)	0.107		
LDH (U/L) at first-line ARSI	≥192 vs. <192	1.58 (0.79, 3.31)	0.200		
ALP (U/L) at first-line ARSI	≥271 vs. <271	1.87 (0.89, 3.75)	0.097		NS
Time to CRPC (months)	≤54 vs. >54	1.78 (0.82, 4.43)	0.172		
PSA (ng/mL) at initiation of ARSI	>3.67 vs. ≤3.67	2.93 (1.46, 6.26)	0.002	2.93 (1.46, 6.26)	0.002
Previous Docetaxel chemotherapy	Yes vs. No	1.15 (0.28, 3.24)	0.817		
Presence of regional lymph node	Yes vs. No	2.31 (0.59, 8.15)	0.215		
Treatment			0.084		NS
	Surgery vs. CAB	0.47 (0.19, 1.22)	0.118		
	Radiation vs. CAB	1.04 (0.46, 2.59)	0.914		
	Surgery vs. Radiation	2.22 (1.04, 4.92)	0.040		
PSA-DT	≤6 mo vs. >6 mo	1.64 (0.59, 6.82)	0.383		

ALP: alkaline phosphatase; CAB: complete androgen blockade; CI: confidence interval; CRPC: castration-resistant prostate cancer; HR: hazard ratio; LDH: lactate dehydrogenase; mo: month; NS: not significant; PSA: prostate-specific antigen.

## Data Availability

The data presented in this study are available upon request from the corresponding author. Due to privacy considerations and ethical restrictions, the data are not publicly available as they contain information that could compromise the privacy of research participants.

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
