# Peer review of "Clinical Outcomes and Prognostic Factors in Nonmetastatic Castration-Resistant Prostate Cancer Treated with Androgen Receptor Signaling Inhibitors Therapy"

_cancers, 2024, doi:10.3390/cancers16152659_

Round 1

Reviewer 1 Report

Comments and Suggestions for Authors

The manuscript presented by Ryo Fujiwara and co-authors addresses the clinical outcomes and prognostic factors in patients with non-metastatic castration resistant prostate cancer treated with first line androgen receptor signaling inhibitors in Japan. The scientific objective is clearly presented and of relevance in the context of clinical practice.

The manuscript is well written and organized. My comments are related with Figure 1, Figure 4 and Figure 5 legends, which could be re-written for more clarity and accuracy.

Reviewer 2 Report

Comments and Suggestions for Authors

This is a very valuable real-world study of the first-line treatment of nmCRPC. The overall design of this article is ingenious and reasonable, and the conclusions are very reliable. Reliable results have been obtained in terms of MFS, OS and safety.

The discussion section also points out the shortcomings of this article. PSMA-PET can be used as a comprehensive whole-body scan instead of CT and Bone scan. In addition, it is very necessary to increase the number of groups other than ENZ.

Therefore, I recommend accepting this article.

Reviewer 3 Report

Comments and Suggestions for Authors

The authors presented a retrospective review of 127 eligible patients with non-metastatic castration resistant prostate cancer (nmCRPC) treated with first line 13 androgen receptor signaling inhibitors (ARSI) in a real-world setting. The authors reported on overall survival (OS), metastatic-free survival (MFS), and prostate-specific antigen progression-free survival (PSA-PFS).

As many members of this patient population are older, with multiple comorbidities they render them ineligible for clinical trial participation. Therefore, we rely on studies of this nature to provide insight into medication best use.

The study is very useful in that there is a comparison of 4 androgen receptor signaling inhibitors (ARSIs). Many times, studies rely solely on a comparison of enzalutamide and abiraterone.

The study is well performed and well-written with only minor need for grammatical editing.

Specific comments

Methodology: Well-presented and correct for the investigation

              As this was a real-world analysis, inclusion of comorbidity data would have been useful

Results: Concise, thorough, and easy to understand

Discussion: The authors did a good job at referencing pertinent studies and outlining limitations of the present study

Conclusion: The authors did not overreach.

              An indication of next steps would be appreciated
